# Subsequent COVID-19 Prophylaxis in COVID-19 Associated Glomerulopathies

**DOI:** 10.3390/vaccines11071152

**Published:** 2023-06-26

**Authors:** Therese Boyle, Emma O’Lone, Elaine Phua, Janet Anderson, Amanda Mather, Suran L. Fernando

**Affiliations:** 1Department of Clinical Immunology and Allergy, Royal North Shore Hospital, St Leonards, Sydney, NSW 2065, Australia; 2Faculty of Medicine and Health, University of Sydney, Camperdown, Sydney, NSW 2050, Australia; 3Department of Renal Medicine, Royal North Shore Hospital, St Leonards, Sydney, NSW 2065, Australia; 4Immunology Laboratory, Royal North Shore Hospital, St Leonards, Sydney, NSW 2065, Australia

**Keywords:** anti-GBM disease, ANCA-associated vasculitis, COVID-19, vaccination, antibodies

## Abstract

Successful vaccination has been the decisive factor in the overall decline of SARS-CoV2 infection related morbidity and mortality. However, global effects of the COVID-19 pandemic are ongoing, with reports of glomerular disease occurring in relation to both infection and vaccination. A particular rise in anti-GBM disease has been identified. Information is still emerging regarding the optimal management of such cases. We reviewed anti-GBM antibody detection rates at our test center over the past 5 years. We followed three patients with biopsy confirmed glomerular disease temporally related to COVID-19 vaccination. Each patient proceeded to receive subsequent COVID-19 vaccination as per immunologist recommendations. Further assessment included COVID-19 antibody testing in each case. A three-fold increase in significant anti-GBM antibody results noted at our center was associated with COVID infection in 10% of cases, and COVID vaccination in 25% of cases. We demonstrated that subsequent vaccination did not appear to lead to adverse effects including relapse in our three cases of COVID-19 vaccine-associated GN. We also identified positive COVID-19 antibody levels in two out of three cases, despite immunosuppression. We report a rise in anti-GBM antibody disease incidence. Our small study suggests that COVID-19 antibody testing can help determine COVID prophylaxis requirements, and subsequent vaccination with an alternative vaccine type appears safe.

## 1. Introduction

Glomerulopathies encompass a spectrum of nephritic and nephrotic renal disease including anti-glomerular basement membrane (GBM) disease, minimal change disease (MCD), and anti-neutrophil cytoplasmic antibody (ANCA)-associated vasculitis (AAV)/glomerulonephritis (GN) [1,2]. Environmental factors such as infection and vaccinations are well recognized triggers of glomerular disease in genetically predisposed individuals [3,4,5,6]. There have been increasing case reports of GN including anti-GBM disease following COVID-19 infection and COVID-19 vaccination. The literature suggests a nearly 70% increase in incidence of anti-GBM disease [7] (p. 180), of which at least 50% of cases were likely due to COVID-19 infection [8,9]. 

Immune mediated glomerulopathies are associated with renal failure and severe morbidity. Ongoing vaccination against COVID-19 is recommended and yet the safety of subsequent vaccination in this group of COVID-19 associated GN is largely unknown. Furthermore, the treatment of immune-mediated glomerulopathies usually requires prolonged potent immunosuppression; [10] (p. 876) the efficacy of further vaccination in this immunosuppressed group is unclear with a paucity of data on the correlates of protection following subsequent vaccination. 

## 2. Materials and Methods

We performed a retrospective review of positive anti-GBM antibody titers detected by our laboratory over the past 5 years (January 2018 to December 2022) to determine incidence. The positive cut-off for anti-GBM antibody results is >20 U/mL with low level antibody titers defined as <60 U/mL [11,12]. We also correlated positive serum samples with subsequent biopsy confirmation of anti-GBM disease. Significant anti-GBM results were defined as those occurring in the context of biopsy confirmed disease and/or deemed as clinically significant by the clinician. We compared the incidence of significant anti-GBM antibody titers before and after the onset of the COVID-19 pandemic. We also reviewed these patients’ details regarding associated COVID-19 infection and vaccination, which we defined as that occurring within 6 weeks of GN disease onset. 

We also performed a prospective study to look at safety of subsequent COVID-19 vaccination in 3 patients found to have a COVID vaccine associated GN over a 3-month recruitment period. Two patients developed anti-GBM disease following COVID-19 vaccination, published in a separate case series [13]. The third patient developed minimal change nephropathy post COVID-19 vaccination. All patients had biopsy confirmed glomerular disease temporally related to COVID-19 vaccination over the 3-month period. Each of these patients consented to participate in a prospective follow-up study. All patients elected to receive the Australian Technical Advisory Group on Immunization (ATAGI) recommended course of COVID-19 vaccinations for those receiving potent immunosuppression at the specified time of the study. If the primary course of 2 vaccinations was incomplete these patients received a total of 3 primary vaccinations followed by booster doses [14]. An alternate vaccine with a different mode of action to that associated with disease onset, was selected for each patient.

Each 4 week follow up study visit consisted of a clinical assessment, laboratory investigations (FBC, UEC, CRP, ESR, anti-GBM antibody level, anti-MPO antibody level, ANCA titer, and urinalysis for blood and protein), as well as review of immunomodulatory therapy. In this report we highlight results at presentation, prior to subsequent vaccination, and 1 month after the subsequent re-vaccination series. Anti-GBM antibodies were measured by a fluorescence enzyme immunoassay (FEIA, ThermoFisher, Uppsala, Sweden). Anti-MPO antibodies were measured by a chemiluminescent assay (BIO-FLASH, Werfen, Barcelona, Spain). We analyzed serum samples from these 3 patients for the presence of anti-spike and anti-nucleocapsid antibodies following each subsequent vaccination. Testing was performed in duplicate by our research based COVID-19 antibody assay (Australian quantitative anti-IgG ELISA (CELISA) manufactured by Cellabs, Sydney, NSW, Australia), using a recombinant trimeric spike protein to measure antibodies to the viral spike and nucleocapsid proteins variants of concern. Results greater than 50 BAU/mL are considered positive, and extrapolation of the upper limit via Graphpad prism software enables measurement greater than the otherwise upper limit of 1000 BAU/mL. The assay has high performance characteristics being >90% sensitive and specific for both anti-spike and anti-nucleocapsid antibodies. 

Exploratory analysis was performed using SPSS statistical software to provide a descriptive statistical summary of results. 

## 3. Results

### 3.1. Anti-GBM Antibody Results

During an audit of serum anti-GBM results in our laboratory from January 2018 to December 2022, 1800 samples were tested, 300 of which were known cases of anti-GBM disease or had previously tested positive prior to January 2018. There were 24 new positive anti-GBM antibody results of which 20 had confirmed anti-GBM disease.

Notably 15 (75%) of these new anti-GBM disease cases occurred between December 2019 and December 2022, during the first two years of the COVID-19 pandemic in Australia. We reviewed the details of these 15 cases. Three developed anti-GBM disease prior to the COVID-19 vaccine roll-out in March 2021 and did not have a prior history of COVID-19 infection. Two of the patients are described in our case series. Two patients received a BNT162B2 mRNA COVID-19 vaccination within 2 months of the onset of disease with one developing symptoms four weeks after a third BNT162B2 mRNA vaccination, and the other 6 weeks after a fourth COVID-19 vaccination (2 × *ChAdOx1* nCoV-19 then 2 × BNT162B2 mRNA). Both of these patients presented following the recruitment period of our study. Two of the remaining eight patients tested positive for COVID-19 infection on presentation of their anti-GBM disease with one having received a COVID-19 vaccine 3 months previously. One patient did not have a history of previous COVID-19 vaccination or COVID-19 infection. All other patients received a COVID-19 vaccination at least 4 months prior to disease onset.

We noted an increase in the number of cases of positive anti-GBM antibody results detected in our laboratory since 2018, with an almost three-fold increase since the onset of the COVID-19 pandemic. We also noted a significant proportion of cases consistently occur in the second half of the year. There was a positive correlation between the increased number of positive anti-GBM antibody results and biopsy confirmed anti-GBM disease. The percentage of new positive serum anti-GBM antibody results in 2022 was 2.1% as compared to 0.6% in 2018, respectively (*p* < 0.05) (Figure 1).

### 3.2. Subsequent Vaccination Safety in Patients with COVID-19 Vaccine-Associated GN

Three patients, two female and one male with a mean age of 67 years, presented to a tertiary referral center. 

Patient A had a history of hypertension, and bowel cancer with hemicolectomy in remission. One month after receiving a second *ChAdOx1* nCoV-19 vaccination patient A presented with a 2-week history of diarrhea and malaise. 

Patient B presented 6 weeks after a first COVID-19 (*ChAdOx1* nCoV-19) vaccination with diarrhea, myalgia’s, and lethargy on a background medical history of hypertension, coronary artery disease, and mitral regurgitation. 

Patient C had no prior medical history and presented with 2 days of lower limb oedema and shortness of breath after their first COVID-19 (BNT162B2 mRNA) vaccination. 

All patients presented with acute kidney injury and proteinuria as defined by KDIGO [15]. ANCA was indeterminate in A and B due to a low-level ANA. Patient A and B both had anti-MPO antibodies detected along with high titer anti-GBM antibodies. Additional autoimmune serology including ENA and anti-dsDNA antibodies was negative in all cases. All three patients had isolated renal involvement with biopsy confirmation of disease. Two patients (A and B) had biopsy confirmed anti-GBM renal disease, both with at least 90% glomerular crescents. Patient (C) had minimal change nephropathy on renal biopsy with 80% podocyte effacement. No patient had a suggestive clinical history, positive lateral flow, or PCR test for SARS-CoV-2 infection. Other infection was also excluded in each case prior to treatment commencement (Figure 2).

Both patients A and B received hemodialysis, induction plasma exchange, oral cyclophosphamide 100 mg, pulse intravenous methylprednisolone 1 g × 3, followed by a tapering course oral prednisone from 1 mg/kg. Patient A was switched to rituximab due to disease severity as well as neutropenia on cyclophosphamide. Patient C was treated with an ACE inhibitor and 1 mg/kg of oral prednisone. Treatment regimens were determined by recommended guidelines such as KDIGO [16] (Figure 2).

Following remission, further vaccination against COVID-19 was administered to each patient (Figure 2 and Figure 3). Patient A remained on rituximab therapy during subsequent vaccination. Patient B had successfully completed the immunosuppression course with stable ANCA titers, levels of anti-MPO antibodies and anti-GBM antibodies prior to subsequent vaccination. Patient C achieved complete remission; reduction in proteinuria to <0.3 g/d or PCR < 30 mg/mmol, stable serum creatinine (~80 umol/L), and serum albumin > 35 g/L [13].

All patients tolerated a further two vaccinations without adverse events. On follow up post revaccination, patient A and B continued to have stable creatinine ~800 umol/L, ANCA titers, levels of anti-MPO antibodies or anti-GBM antibodies and normal CRP, with no clinical features of disease flares. Neither patient A nor B required any GN treatment modification. However, patient A did receive tixagevimab and cilgavimab (Evusheld) prophylaxis as recommended at the time for patients expected to have a poor response to vaccination [17,18]. Patient C experienced a disease flare noted by fluid retention/3 kg weight gain, hypoalbuminemia (24 g/L), and a rise in UPCR to >300 mg/mmol, 6 weeks following *Novavax* (NVX-CoV2373) COVID-19 vaccination and 1 week following the onset of mild COVID-19 infection. The patient achieved remission following initiation of tacrolimus and further tapered course of high dose steroid therapy (Figure 2).

The results of anti-COVID-19 antibody testing showed the development of a positive anti-spike antibody titer in two of the three patients at different time points (Figure 3). Patient A had an anti-spike antibody level below the positive cut-off of 50 BAU/mL despite four vaccinations. Patient B had a positive anti-spike antibody result on initial testing following two primary vaccinations which doubled subsequent to the third primary vaccination and increased by approximately 25% following the first booster vaccination. Patient C only developed a positive anti-spike antibody titer after his first booster vaccination, i.e., fourth vaccination. No patient developed positive anti-nucleocapsid IgG antibodies on final measurement reflecting absence of natural immunity from COVID-19 infection. 

## 4. Discussion

COVID-19 vaccination programs are an integral part of a worldwide strategy to curb the spread of COVID-19 infection and reduce severity of disease. We have demonstrated a rise in GN associated with COVID-19 infection and vaccination in our population which confirms findings in the literature. Our study also suggests that it is safe for patients who have developed COVID-19 vaccine-associated GN to receive subsequent vaccination with a different class of vaccine. The efficacy of such vaccination can be further assessed via measurement of correlates of protection. 

Despite the association of glomerulopathy with vaccination in our patients, the increased risk of severe disease and death following COVID-19 infection due to renal disease and potent immunosuppression, warranted consideration of further vaccination [19]. In contrast to immediate allergic reactions, there is no standardized testing available to determine safety of subsequent vaccination for patients with autoimmune GN [20]. Whilst others have utilized the same vaccine type [21] (p. 2), our patients each received an alternative COVID-19 vaccine to that associated with their glomerulopathy. A significant finding of our study is the safety of further vaccination. Subsequent vaccinations did not appear to lead to an increase in anti-GBM antibodies, ANCA titers or a flare in minimal change disease in our small case series. Although two patients (A and B) were receiving hemodialysis, neither developed additional organ involvement suggesting disease relapse as per BVAS 2003 [22,23]. Patient C’s flare of disease followed and therefore was more likely associated with active COVID-19 infection although the effect of the NVX-CoV2373 vaccine administered 6 weeks previously cannot be completely excluded [24]. Canney et al. recently showed no significant relapses in patients with glomerulopathy after an initial COVID-19 vaccination, although a two-fold increase in the relapse rate of disease was observed following subsequent vaccination (of the same mechanism in 97%) [25]. However, their study did not include patients with anti-GBM disease. Concomitant immunosuppressive treatment during subsequent vaccination may protect against autoimmune/glomerular disease, although relapses have still been reported early in the course of therapy [26]. The treatment outcome of COVID-19 vaccine-associated GN remains unclear but one study suggests that anti-GBM disease may respond less favorably to immunosuppression than de novo disease [26]. 

COVID-19 vaccination leads to protection against subsequent severe acute respiratory syndrome coronavirus 2 (SARS-CoV-2) infection via induction of antibodies to the viral surface spike (S) protein. Natural infection also leads to development of these antibodies as well as antibodies to the nucleocapsid protein (NP) antigen [27,28]. Whilst the detection of SARS-CoV-2-specific antibodies by our method does not have diagnostic clinical utility, the results may represent a correlate of protection from subsequent infection. For instance, Feng et al. found that anti-spike antibody levels > 264 BAU/mL were effective against symptomatic COVID-19 infection 80% of the time. A possible explanation for the reduced vaccine response by patients A and C may be the initially high levels of proteinuria (>150 mg urinary protein/day) as well as secondary hypogammaglobulinaemia acquired from the immunosuppressive treatment both patients received. Patient A had complications of stage 3 acute kidney injury on presentation as well as high dual anti-MPO and anti-GBM antibody levels requiring more aggressive treatment, which may have blunted the vaccine response [29,30,31,32]. Further, it is recognized that COVID-19 booster vaccination may only lead to seroconversion in one third of patients on anti-CD20+ monoclonal antibody treatment [32,33,34,35]. Whilst the NVX-CoV2373 vaccination resulted in a significant anti-spike antibody response for patient C, the timing of anti-COVID-19 antibody testing did not permit the adoption of a similar strategy for patient A based on her lack of response to BNT162B2 mRNA vaccination. 

It is particularly important to recognize vaccine-associated GN as a clinical entity, and have awareness of management strategies due to emergence of case reports and series following the implementation of COVID-19 vaccination programs [3,6,8,36,37,38]. Recent systematic reviews of renal disease associated with COVID-19 vaccination showed that MCD occurred most frequently, followed by IgA nephropathy, and vasculitis [5,39]. An increasing prevalence rate may have been evident preceding the COVID-19 pandemic; however, our laboratory has reported a higher-than-expected rise in positive anti-GBM antibody results with biopsy confirmation of disease between 2018 and 2022. This is the first study in the Australian population to echo findings in the international literature. Our additional cases were temporally related to COVID-19 vaccination, and neither patient had evidence of preceding or concurrent COVID-19 infection supporting, although not proving, vaccine causation as demonstrated in other reports [40]. Whilst almost 90% of vaccine associated renal disease cases have occurred after mRNA vaccination, notably two out of our three cases followed administration of the *ChAdOx1* nCoV-19 viral vector vaccination [3,5,40,41,42]. The temporal relationship between vaccination and the onset of GN varies in the literature, consistent with our cases; 4 weeks is most frequently reported but studies have also reported cases with onset after 3 months [43]. Specific timing of likelihood of causality has been proposed for different COVID-19 vaccine associated conditions, with probable causality of cases occurring within 6 weeks, and causality unlikely after 12 weeks post vaccination [44]. Notably such divisions are not definitive or generalizable [45] (p. 2), and to our knowledge are not available for vaccine associated glomerulopathies. Anti-GBM disease has a bimodal peak age distribution; more commonly occurring in younger males (20–30 years) and older females (60–70 years) [46,47]. There was hence no significant difference regarding the age of presentation in our small case series including COVID-19 vaccine associated anti-GBM disease. 

It is important to recognize that cases of anti-GBM disease and MCD are not isolated to vaccination but have also been seen in association with COVID-19 infection [3,4,21,48]. COVID-19 infection has most commonly been associated with collapsing glomerulopathy, although other types of immune mediated renal disease have been reported including AAV [49]. The rise in cases of anti-GBM disease since the onset of the COVID-19 pandemic suggests that an infective or vaccination trigger is involved in the pathogenesis in a proportion of cases of anti-GBM disease [3,8]. 

Current evidence suggests that following either infection or vaccination, antigens are recognized as potential pathogens by innate immune system pathogen-associated molecular patterns (PAMPs) or damage-associated molecular patterns (DAMPs), and receptors on immune cells including pattern-recognition and Toll-like receptors (TLR). This leads to transcription of target genes, with resulting release of pyrogenic cytokines (i.e., interleukin [IL]-1, IL-6, tumor necrosis factor-alpha, and prostaglandin-E2). A subsequent cascade of events ensues including phagocytosis, further release of inflammatory mediators including chemokines and cytokines, and complement activation. Notably, unlike in natural infection, the current mRNA vaccines are modified to reduce binding to TLRs and immune sensors, thus limiting production of cytokines such as type I interferon [50,51,52,53]. Meanwhile, other cytokines such as BAFF subsequently stimulate adaptive T and B cells leading to antibody production. The response to infection causes inflammation at the target sites of the invading virus. Conversely, the immune response to vaccination is targeted to an attenuated virus or antigen, which for most COVID-19 vaccines is the viral spike (S) protein [54]. Further intricacies of the specific immune response varies according to the type of infection and/or vaccine [55]. In both instances the elimination of pathological microbes and proteins must be controlled so to avoid responses that produce excessive damage of self-tissues. An aberrant and dysregulated immune response typically in the setting of reduced regulatory T cells could be triggered by infection or vaccination, and can lead to production of pathological autoantibodies such as anti-GBM antibodies [56]. Autoantibodies to endogenous antigens via either molecular mimicry or antigenic injury, the release of proinflammatory mediators including cytokines, adjuvants in vaccines to augment the inflammatory response, and direct viral cytopathy with SARS-CoV-2 infection may all play a role in this process [57,58,59]. Whilst it is unclear, the mechanism of the vaccine type may have a role in such autoimmune disease pathogenesis, for instance stronger priming of CD8+ T cell responses by self-amplifying lipid encapsulated mRNA vaccination compared to adenoviral vector vaccines [55].

The increased anti-GBM antibody detection rate in the second half of the calendar year is an interesting observation and perhaps reflects the onset of disease following the southern hemisphere winter viral peak [60]. This increase may not occur with other types of vaccine associated glomerular disease [61]. Aside from COVID-19 infection and vaccination, alternative possible factors involved in this seasonal variation include other infections and vaccinations during the winter period [60]. In terms of the rise coinciding with spring; it is known that allergic disease including seasonal allergic rhinitis increases the risk of autoimmune conditions and glomerular disease, but this association has not been established for anti-GBM disease [62,63].

Our study has limitations. Such a small case series means we can only describe these interesting results and are unable to draw any concrete conclusions regarding safety or efficacy of the vaccines, or indeed causation of GN secondary to COVID-19 vaccinations. Furthermore, our antibody testing did not include a confirmatory neutralization step to confirm protection and thus only provides a correlate. Anti-nucleocapsid antibody levels may wean quickly, making timing of testing important, particularly in the absence of simultaneous COVID-19 PCR testing. However, this small study does contribute to the growing number of case reports in the literature of COVID-19 vaccine induced GN and supports the potential for considering repeated vaccines against COVID-19 but using an alternative vaccine to that which induced the GN. Larger longitudinal studies are required for further confirmation. Further, all adverse events following vaccination (AEFI) should be reported, and national vaccine safety surveillance programs such as AusVaxSafety and Vaccine Adverse Event Reporting System (VAERS) can help determine causation and assist regulatory agencies and advisory groups with subsequent vaccination recommendations.

## 5. Conclusions

To our knowledge, this is the first demonstration in an Australian population of an increased prevalence of anti-GBM disease over the time of the COVID-19 pandemic. We also concur with international findings of an increasing rate of anti-GBM disease since the onset of the COVID-19 pandemic with post COVID-19 vaccination cases contributing to the disease burden. 

Whilst a small case series, this is also the first literature concerning safety and efficacy of alternate vaccination in patients with COVID-19 vaccine-associated GN. Correlates of protection can assist with determining if natural infection has occurred in such patients, and with identifying patients who may benefit from alternative COVID-19 prophylaxis due to an insufficient anti-spike antibody response. We recommend using an alternative vaccination type for subsequent vaccination of patients with COVID-19 vaccine-associated glomerular disease. However, confirmation regarding safety of further vaccination requires large multicenter longitudinal studies. Meanwhile, the outcomes from our study remain encouraging and show that further vaccination can be considered. 

## Figures and Tables

**Figure 1 vaccines-11-01152-f001:**
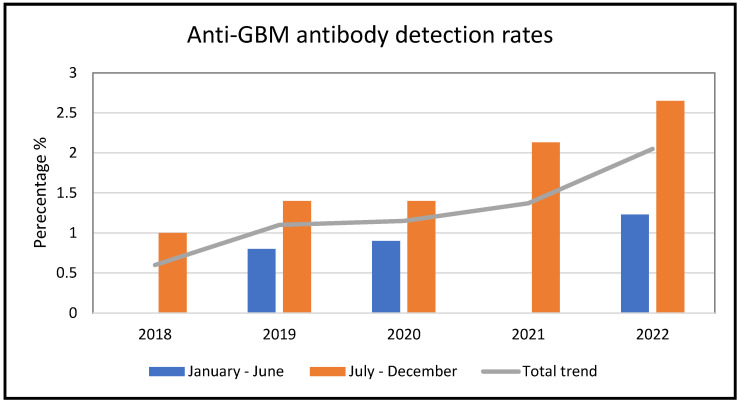
Graph showing % of new significant positive anti-GBM antibody titers according to the total number of samples tested between January 2018 and December 2022 at a tertiary referral laboratory in Sydney, Australia.

**Figure 2 vaccines-11-01152-f002:**
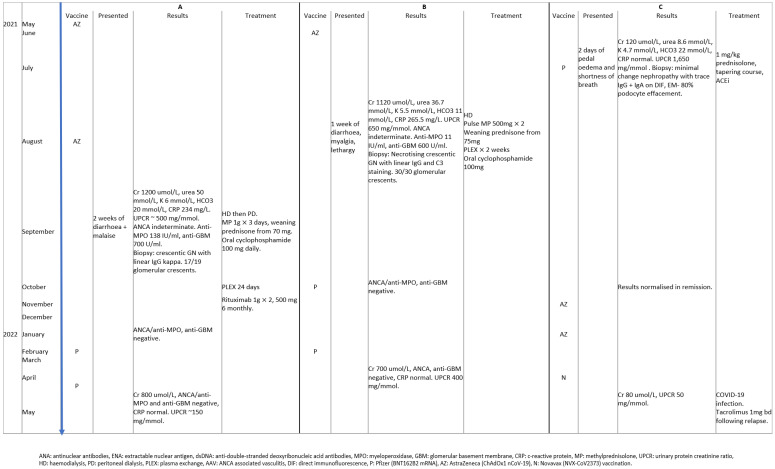
Timeline of the characteristics of the three patients (A–C) with glomerulopathies following COVID-19 vaccination.

**Figure 3 vaccines-11-01152-f003:**
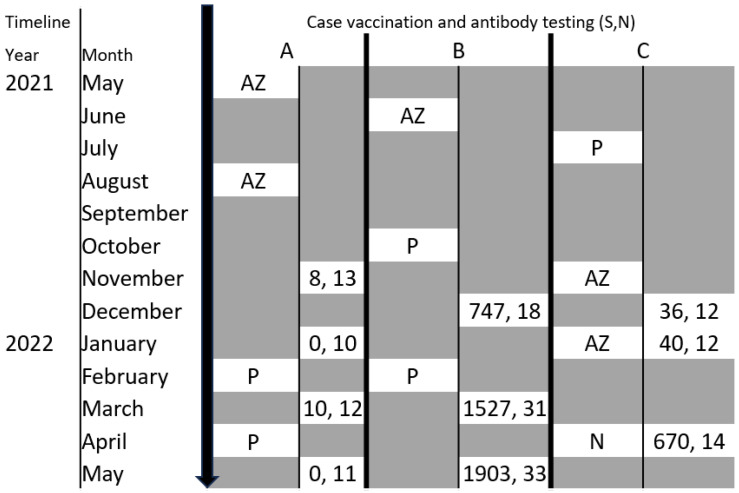
A timeline of the serological evaluation of anti-SARS-CoV-2 antibodies in study patients following vaccination. S: Anti-spike binding antibodies (BAU/mL) (positive > 50 BAU/mL), N: Anti-nucleocapsid antibodies (BAU/mL) (positive > 50 BAU/mL) AZ: AstraZeneca (*ChAdOx1* nCoV-19) vaccination, P: Pfizer (BNT162B2 mRNA) vaccination, N: Nuvaxovoid/Novovax (NVX-CoV2373) vaccination.

## Data Availability

Data are contained within the article.

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
