# Peer review of "Subsequent COVID-19 Prophylaxis in COVID-19 Associated Glomerulopathies"

_vaccines, 2023, doi:10.3390/vaccines11071152_

Round 1
Reviewer 1 Report
The authors describe their experience with vaccination for COVID-19 following diagnosis of COVID-19-associated glomerulopathies.
This is a case series of 3 cases, which is rather limited to extract safe conclusion.
The presentation of the cases could be improved and described separately.
Follow-up data after the re-vaccianation.
The indication of monoclonal antibodies in these cases could be also commented.
ok
Reviewer 2 Report
Subsequent COVID-19 prophylaxis in COVID-19 associated glomerulopathies.
Interesting retropective paper that reports 3 fold increase of significant anti-GBM antibodies related to COVID: infection in 10% of cases and vaccination in 25% of cases in a single center in a 5y period.
3 patients had biopsy confirmed glomerular disease ( 2 anti-GBM and 1 minimal disease) over a 3 month period after COVID-19 vaccination. These patients were subsequentelly re-vaccinated for COVID with another vaccine type.
Couple considerations for publication:
The three patients you describe had +60y. I consider it is very important to compare it with the average age that patients develop Anti-GBM and minimal disease, so that the reader have a reference.
You should add in Table 1 the reference for the laboratory values you cite. This can have a huge variation across the globe and it might help readers.
I suggest table 1 and 2 to be converted in a time line. It will get more clear than with the Tables.
It is very important to call atention that, patient´s COVID seroconvertion is considerably reduced with Rituximab. This is already well described in the literature and this might be the case in patients 1 especially considering that she also got cyclophosphamide.
3 patients are too litlle to come to any conclusions regarding safety COVID vaccination. This has been described in the paper by the authors by I consider it should be reinforced.
Reviewer 3 Report
This is a retrospective investigation of COVID-19 vaccines in anti-GBM positive patients and case series of those vaccine-associated glomerulopathy. Authors demonstrated an increased prevalence of anti-GBM disease over the time of the COVID-19 pandemic. Moreover, in case series, subsequent vaccination with an alternative vaccine formulation was safe. Therefore, authors recommended to be aware of the development of anti-GBM disease when using the COVID-19 vaccine and to use alternative vaccine types in the event of vaccine-associated glomerular disease.
The presented study was well performed, and the manuscript was described in a reasonable manner. Although only anti-GBM disease was investigated in the retrospective cohort, the case other than anti-GBM disease was included in the case series. I think it would be better to limit case reports to anti-GBM disease (exclude case C) or to unify the subject of cohort studies to all renal diseases.
Moreover, is it correct that 11 of the 15 cases developed anti-GBM disease after COVID-19 vaccination, and two cases after COVID-19 infection? I think it would be more useful if authors could provide more details about the relationship between the COVID-19 vaccine and the onset of the anti-GBM disease in those 11 cases rather than the detailed results in the case series of only two cases.
Finally, what do authors think is the reason for the higher incidence of anti-GBM disease in the second half of the year?
Round 2
Reviewer 3 Report
This is a retrospective investigation of COVID-19 vaccines in anti-GBM positive patients and case series of those vaccine-associated glomerulopathy. The presented study was well performed, and the revised manuscript is described in a reasonable manner. Authors had also responded to my all comments.